# Hydroculture Cultivation of Strawberries as Potential Reference Material for Microcystin Analysis: Approaches and Pitfalls

**DOI:** 10.3390/toxins17060285

**Published:** 2025-06-06

**Authors:** Wannes Hugo R. Van Hassel, Benoît Guillaume, Julien Masquelier

**Affiliations:** Organic Contaminants and Additives, 1050 Sciensano, Belgium; benoit.guillaume@sciensano.be (B.G.); julien.masquelier@sciensano.be (J.M.)

**Keywords:** hydroculture, microcystins, plants, reference material, UHPLC-MS/MS

## Abstract

Toxic cyanobacterial blooms are prevalent in surface waters. Depending on several conditions, these blooms produce cyanotoxins. Human exposure to these toxins can occur through multiple routes, including contaminated crops through irrigation with contaminated water. Analytical methods have been developed for cyanotoxin quantification to investigate these exposures. Yet, proper comparisons between different labs via proficiency tests or interlaboratory comparison tests, as well as method quality assurance, are impossible. Developing reference materials for cyanotoxins in plants would help resolve these problems. Therefore, a novel liquid hydroculture setup was optimized to grow and contaminate strawberries. During fruit ripening, these plants were exposed to growth medium contaminated with pure microcystin-LR or freeze-dried cyanobacterial biomass containing different microcystin congeners. Afterwards, fruits, greens, and roots were harvested. Validated UHPLC-MS/MS methods were used to quantify the microcystin congeners in the growth medium and the plants. Furthermore, both contamination conditions resulted in the accumulation of toxin(s) in the roots and the greens. Yet in the contamination models, no toxin(s) accumulated in the fruits. Therefore, this contamination approach, combined with strawberries as a berry plant model, is only suitable for reference material production for limited matrices. Our cultivation model to produce reference material could be evaluated for other berry producers.

## 1. Introduction

The prevalence of toxic cyanobacterial blooms worldwide is, by now, a well-established fact [1]. Their prevalence could be a growing burden on public health, considering that climate change is increasing frequencies and prolonging bloom occurrences in the previously more temperate parts of the globe [2,3,4,5,6,7].

Common toxins produced during these blooms can be divided into neuro- and hepatotoxins. A diversity of microcystin congeners (MCs) are the most characterized toxin group of hepatotoxins [8]. These toxins, especially microcystin-LR (MC-LR), are commonly observed in freshwater blooms. Next, the structurally related hepatotoxin Nodularin (NOD) is more common in brackish/saltwater blooms [9,10,11]. The last member of the hepatotoxin group is cylindrospermopsin (CYN) [12]. This alkaloid was initially discovered in *Cylindrospermopsis raciborskii* [13]. The neurotoxin group includes anatoxin-a (ATX) variants and saxitoxin (STX) analogs. Both groups can cause neurological disorders and even death upon acute exposure [14,15].

Direct human exposure to cyanotoxins through water, through either consumption or recreation, is the most evident [8,16]. Therefore, the World Health Organization has proposed tolerable daily intake values for MC-LR equivalence (0.04 µg g_bodyweight_^−1^ day^−1^) as the total for all MCs [8]. Moreover, multiple guideline values have been proposed for lifetime and/or short exposure to MCs, ATX, STX, and CYN through drinking water or recreation [8,14,15,16].

However, secondary exposure through cyanotoxin accumulation in different foodstuffs is also a potential scenario [7]. The consumption of aquatic animals (fish, shrimp, snails, etc.), algae-based food supplements, or products from crops irrigated with contaminated water, are shown as potential pathways for cyanotoxin exposure [17,18,19,20,21,22,23,24,25,26,27,28,29,30,31,32,33].

Specifically for plants, the accumulation of MCs in different crops has been confirmed in multiple studies under lab, field, and mid-range greenhouse conditions [23,24,25,34,35,36,37,38,39,40,41,42], while limited research on CYN accumulation has been performed [43,44,45]. Most research has focused on soil-grown plants using spray or drip irrigation.

Many properly validated analytical methods are available to quantify cyanotoxins in diverse matrices to mediate and study the risk of exposure from potential exposure pathways [18,20,43,46,47,48,49,50,51,52,53,54,55]. Yet continuous inter-lab comparisons of these methods is still impossible for most applications. Such comparisons are usually accomplished by organizing proficiency tests using well-defined contaminated material, preferably certified reference material (CRM), that adheres to ISO 17034 [56]. However, using reference material (RM) is also acceptable for these comparisons. RMs generally lack certain levels of characterization and documentation compared to CRMs. These characterizations of CRMs include extensive stability and measurement uncertainty studies and the determination of assigned values for toxin concentrations based on analysis in multiple labs, which are associated with a high cost.

CRMs are currently only available for microcystins and cylindrospermopsin-lyophilized water blooms [57]. Moreover, feasibility studies showed the potential to produce CRMs for anatoxin-a, microcystin congeners, and cylindrospermopsin in lyophilized algal and mussel tissue [58,59]. While these studies already show the potential of producing CRMs in different matrices using lyophilized cultures, research on other matrices, like dedicated fruit and vegetables, should still be tackled. Moreover, consistent interlaboratory comparison tests or proficiency tests (PTs) are lacking for most matrices. Currently, only proficiency tests for drinking and recreational water are organized yearly. Moreover, these PTs use RMs that are currently not ISO 17034-certified. Organizing PTs in diverse matrices with the right certification will be vital to increase the reliability of analytical results for cyanotoxin analysis throughout the world. Furthermore, improving the reliability of the available analytical approaches will affirm current and support future legislation on regulatory limits for cyanotoxins. Additionally, increasing the available RMs should speed up the adoption of current analytical methods and the development of new ones.

As a first step in potentially producing reference materials for crop plants, our current study evaluated the use of hydroculture to contaminate strawberries with MCs originating from freeze-dried bloom material. This kind of setup could be very advantageous as hydroculture setups eliminate the retention of cyanotoxins in soil, resulting in improved control of toxin concentration and reducing the quantity of toxin needed to contaminate the plants. Moreover, the amount of dangerous biological waste is reduced. The plant cultivation setup was evaluated. Also, the MC concentrations were quantified using Ultra-High-Performance Liquid Chromatography–tandem mass spectrometry (UHPLC-MS/MS).

## 2. Results

### 2.1. Cultivation Approach and Harvest Measurements

The successful cultivation of the strawberries was dependent on the tight control of root submersion in the Hoagland solution. As hinted at in the method description, about half of the plants had their roots submersed completely during the cultivation and contamination. For the other plants, the volume of Hoagland solution was kept at half the length of the roots. The latter resulted in the best-performing cultivation of strawberries, with 73.0% of plants properly growing after the start of the experiment. The other 27.0% of the plants had difficulty forming roots, which resulted in a lack of stem and leaf development. For the plants with submerged roots, the development of stems, leaves, and flowers seemed to pose no problem during the initial growth phase. However, the leaves started showing discoloration at the edges after 3 weeks. Moreover, a pungent smell started emanating from the Hoagland solution, caused by root rot. Thereafter, the strawberry plants withered within the span of 1 week. This clearly shows the need to control Hoagland solution levels strictly (with half-submerged roots) to grow strawberry plants in hydroculture effectively. After the onset of fruits, the surviving plants were divided into the blank, MC-LR, and NM conditions for four weeks. The weight of the different plant parts (e.g., greens, roots, and strawberries) was measured after harvesting (Figure 1). Using a Wilcoxon test, no significant difference was measured for the same plant parts between the different conditions (*p* < 0.05).

### 2.2. Microcystin Congeners Quantification in Hoagland and Accumulation in Plant Material

As earlier described, the theoretical fortified concentration of total analyzed MCs in the Hoagland solution should have been 100 µg L^−1^. This concentration was successfully obtained with an average concentration of 100.41 µg L^−1^ with a standard deviation of 27.02 µg L^−1^ (Figure 2). About half of this standard deviation can be explained by the measurement uncertainty (MU) of the analytical method, ±14 µg L^−1^. An additional source of variation might have resulted from the freeze-dried biomass not being completely homogeneous. The standard variation in the observed total MC concentrations in the freeze-dried biomass was ±8.7 µg g^−1^. Moreover, additional elements, like the weighting of the biomass, can contribute to increases in the standard deviation.

On average, MC-RR, MC-LR, and MC-YR were the most prevalent congeners at 31.5%, 37.3%, and 25.8% in consecutive order, compared to the total analyzed MCs in the Hoagland samples. Additionally, small amounts of MC-WR, MC-LA, MC-LY, MC-LF, and MC-LW could be detected or quantified (Appendix A). A complete overview of the data obtained for MCs in Hoagland solution can be found in Appendix A.

For the MC-LR-fortified Hoagland solution, the measured concentration did not correspond to the theoretical 100 µg L^−1^ (Figure 2). Instead, an average value of 15 µg L^−1^ MC-LR was measured. However, these low concentrations could have been caused by the instability of MC-LR in the Hoagland solution during storage at −20 °C. A preliminary experiment, unpublished up to now, in 2022 showed that 80.82 µg L^−1^ (MU = ±16.28 µg L^−1^) could be observed directly after fortifying Hoagland solution with pure MC-LR (timepoint 0) at a theoretical concentration of 100 µg L^−1^. During this experiment, Hoagland solution fortification was performed in triplicate to show the stability of the MC-LR in solution during 6 days of storage in a fridge or at room temperature. After 6 days of storage, 83.41 µg L^−1^ (MU = ±16.80 µg L^−1^) and 82.70 µg L^−1^ (MU = ±16.6 µg L^−1^) were observed for the refrigerated and room-temperature conditions, showing no degradation of the MC-LR. This suggests that the MC-LR concentration in Hoagland solution during the experiment would have been close to the theoretical fortified concentration.

The accumulation of MCs in the plants was observed for the roots and the greens (Figure 3 and Figure 4). Independent of the contamination source (pure MC-LR or NM), no MCs were detected in the strawberries. The highest total analyzed MC concentrations were found in the root samples.

The average total concentration for the natural microcystin congener mix-treated roots was 417.1 ng g^−1^ (Figure 3). All these samples contained a mixture of MC-RR, MC-LR, MC-YR, and MC-LW (Appendix A), similarly to the Hoagland solution used for this condition. A complete overview of the data obtained for MCs in the root samples can be found in Appendix A. MC-RR contributed, on average, 50% to the total analyzed MC concentration. Multiple samples also contained different mixtures of MC-LA, MC-LF, MC-LY, and/or MC-LW. Their concentrations were, on average, lower than 1.5 ng g^−1^ and thus only minimally contributed to the total MC concentration. For MC-LR-treated roots, the average MC-LR concentration in the roots was 370.1 ng g^−1^. There was no significant difference between the accumulated concentration of MC-LR and the total concentration of MCs for the NM condition using the Wilcoxon test statistic at *p* < 0.05.

For the natural microcystin congener mix-treated plants, on average, 3.4 ng g^−1^ of total microcystin was observed in the greens (Figure 4). Primarily, MC-LR and MC-RR were observed in the samples, with minor contributions from MC-YR and MC-LF in some of the samples (Appendix A). A complete overview of the data obtained for MCs the roots samples can be found in Appendix A. For MC-LR-treated plants, an average concentration of 2.8 ng g^−1^ was observed.

## 3. Discussion

During this study, MCs accumulated in strawberry plants during cultivation under pure liquid hydroculture conditions. Finding the ideal volume of nutrient solution for plant growth was challenging. Half-submerging the roots in Hoagland solution was a viable solution. This prevented the roots from developing root rot, for which overexposure to water is a commonly accepted environmental factor. Using a substrate like perlite would be another solution to prevent root rot. Moreover, substrate hydroponics were shown to improve strawberry growth compared to liquid hydroponic setups [60]. However, the main goal of this study was to evaluate the accumulation of MCs in the strawberry plants without potential retention of the toxins on the substrate or soil. The retention of MCs in perlite was already shown for a strawberry hydroculture setup [61]. An additional downside of using substrates in the framework of eventual reference material production is the increased biochemical waste. More precisely, contaminated perlite used during the continuous production of reference material should be removed after use at high cost to conform with biosafety regulations and to limit safety risks for scientific personnel. In comparison, the plants consumed the Hoagland solution with limited residual waste.

During our study, the growth of the strawberry plants under different conditions was not impacted, taking into account the weights of the various plant parts. Yet MC exposure experiments for strawberries in substrate hydroculture setups did show reduced root/shoot length, weight, numbers of leaves, roots/leaves dry and fresh weight, chlorophyll/carotenoid content, and total protein content at concentrations of 10 and 20 µg L^−1^ [39]. However, a different cultivar (e.g., *Fragaria vulgaris* L.) was used during these experiments. Moreover, MC contamination experiments in soil, using spray or drip irrigation for strawberry plants (*Fragaria* var. *Elsanta*), did not result in differences in growth at 100 µg L^−1^ MC-LR compared to the control [34].

During this study MC and MC-LR concentrations were evaluated in the Hoagland solution. MC concentrations in the Hoagland solution were as expected. However, MC-LR concentrations in the Hoagland solution were below the expected value. Similar problems have been observed during other, larger-scale contamination experiments that used pure MC-LR to fortify irrigation water and stored the water samples at −20 °C before analysis [34]. Moreover, the FDA has already observed this kind of instability for MC-LR in water, resulting in the guidelines of keeping water samples refrigerated before analysis, but not frozen [62]. Most likely, several coinciding factors resulted in these reduced MC-LR concentrations during storage. Two well-known factors are the adsorption of the MC-LR to the recipient and the lower pH (5.4–6.3) of the Hoagland solution compared to that of water. Plastic recipients are well known to adsorb MC-LR [63,64]. Additionally, the lower pH increases the hydrophobicity of MC-LR [65]. Moreover, MCs are also known to be less stable at lower pH ranges due to oxidation [66,67]. Multiple oxidants, including CuSO_4_, H_2_MoO_4_, MnSO_4_, and MgSO_4_ are used to make Hoagland solution [68]. MC-LR concentration reduction in water, when spiked with pure standard, is probably a slow process occurring during storage at −20 °C (>30 days). Remarkably, MC degradation does not occur after adding (freeze-dried) cyanobacterial biomass. However, the presence of other organic matter, in this case from the freeze-dried material, in the Hoagland solution is known to reduce the adsorption of MC-LR to the recipient and probably also reduce the overall oxidation of the MCs. Direct analysis of the MC-LR concentration in the Hoagland solution after fortification could not be supported in the framework of this project due to the higher cost (standards, LC-MS/MS operating costs) for analyzing the samples every week.

Besides in the preliminary experiment results, the accumulated concentrations of MCs or pure MC-LR in the different plant parts showed no significant difference between the conditions. Therefore, we are convinced that the plants were exposed to the targeted 100 µg L^−1^ MC-LR concentration during the experiments.

As previously mentioned, both MCs or MC-LR accumulated in different strawberry plant parts (e.g., the roots and the leaves). These results correspond well with earlier obtained results in a soil-based greenhouse setup where strawberries were contaminated with MC-LR-fortified water at similar concentrations using drip irrigation [34]. The observed accumulation differed most in the roots, where our results were 10 times higher, on average. This could be related to the retention of MCs or MC-LR during soil-based cultivation. A soil-based spray irrigation setup and a substrate hydroponic setup also showed lower accumulation of MCs or MC-LR in the roots [34,61]. Interestingly, all four experiments resulted in a similar concentration of MCs or MC-LR accumulated in the stems and leaves of the plant (Table 1).

Spray and substrate hydroponic experiments also showed the accumulation of MCs or MC-LR in strawberries [34,61]. This accumulation was not observed in our experiments or the earlier drip irrigation experiments. The difference between our hydroculture and the substrate setup could be that the contaminated growth solution was sprayed on the substrates during the latter experiment [61]. This could potentially cause the direct contamination of the fruits, similarly to spray irrigation. The observed average concentrations of 1.04 ng g^−1^ and 2.6 ng g^−1^ for substrate hydroponics and spray irrigation, respectively, could also be an indication of this. Yet it should be mentioned that the strawberry plants during the substrate hydroponic approach were contaminated for twice as long compared to those in our experiments, which could potentially explain the eventual MCs contamination in the strawberry fruits [61]. Clearly, the biochemical and kinetic mechanism behind the translocation of MCs into plant tissues should be further investigated. Having MC-contaminated reference materials, including different plant parts, could be helpful to reliably determine MC concentration in plants during this type of fundamental research. In extent, an unresolved knowledge gap relating to the difference in plant accumulation of the chemically diverse MC group should be further investigated. The difference in hydrophobicity, cellular uptake, and metabolic stability should be elucidated, preferably by relevant biochemical and kinetic studies where single MCs are used for contaminations, similar to what was performed earlier for CYN and MC-LR [44].

Overall, our liquid hydroculture model was able to contaminate strawberry plants with MCs and pure MC-LR. It provides a safe, easy-to-use system that also limits the produced waste. While the roots and greens could be used as RM for specific matrices and applications in fundamental research, the usefulness of the cultivation setup is limited for strawberries in the framework of reference material production. The commonly consumed part of the plant, the strawberries, were not contaminated. Other spray irrigation methodologies did result in the low-level contamination of the fruits [34,61]. Yet applying any approach that includes spraying liquids contaminated with toxins should be performed with the utmost care. Safety precautions should be considered to prevent human and environmental exposure to toxin-loaded aerosols produced during spraying. Such precautions could go from personal protection equipment to isolation greenhouses. These precautions would significantly increase the cost of reference material production. Moreover, our current approach could be further optimized to increase yield during a single cultivation and contamination cycle. Indeed, more contaminated material should be produced during a single cycle to effectively organize PTs or use the material as commercial RM. Therefore, dedicated hydroculture systems, like ebb and flow or hydroponic towers, could be investigated. These systems have the benefit that the roots are temporarily submerged completely and thus provide sufficient aeration of the roots [68]. However, the adsorption of MCs to the multiple plastic parts of these kinds of systems should be considered and validated during development. The type of plastic used could have a significant influence on the amount of toxins that can reach the plants [63]. Furthermore, other model organisms should be considered to produce a potential reference matrix, as the strawberries were not contaminated. However, every crop will come with its own challenges for hydroculture implementation. For instance, other berry models are considered shrubs, while a strawberry plant is considered a herb, meaning that adaptations to the cultivation protocol would be required. Therefore, early plant development, nutrient and water requirements, and disease mitigation could be significantly different for other berry plants. Berry plants cultivated using hydroculture setups, like the European blueberry, could be initially selected to implement our experimental setup. A significant hurdle in this case would be the necessity to initially grow the plants in a soil-based culture and transfer them to a suitable hydroculture setup. Besides berry plants, other types of fruiting species, like cucumber, tomato, or similar fruits, could also be selected to implement our experimental setup, as some cultivars are already being grown in hydroculture and widely consumed worldwide. However, it should be mentioned that MC-LR accumulation in tomato was already evaluated during soil cultivation, resulting in no accumulation of MCs in the fruits [69]. On the other hand, a study on pepper plants shows that MC accumulation is possible in the fruits [70]. No such information is available for cucumbers. The overall relevance of these crop plants for food consumption promotes the continued investigation of their suitability as RMs. Furthermore, other plant types, specifically herbs and leafy vegetables, should be considered for RM production as the successful accumulation of MCs and MC-LR was already shown with our hydroculture setup [71].

## 4. Conclusions

Using the optimized liquid hydroculture model for reference material production, strawberry plants were successfully grown, and strawberries were produced. Our approach is safe to use for scientific personnel and reduces produced biochemical waste, like toxin-contaminated soil or substrates. Moreover, both the roots and greens of the strawberry plants were successfully contaminated using either pure MC-LR or MCs originating from dried cyanobacterial biomass, but the strawberry fruits did not accumulate toxins using this procedure. Yet using the contaminated roots and greens as RM could be of limited use for specific matrices and fundamental research. Therefore, more appropriate berry model plants should be investigated that might benefit from our hydroculture setup as a first step towards reference material production. Moreover, other crop types, like leafy vegetables and herbs, could be considered as well to increase the diversity of RM for different fruit and vegetable types.

## 5. Materials and Methods

### 5.1. Chemicals and Standards

The different microcystin congeners used as analytical standards were obtained from Enzo Life Sciences, Bruxelles, Belgium. Acetonitrile and methanol were UPLC-grade (Biosolve, Valkenswaard, The Netherlands). Milli-Q water was produced in-house (conductivity ≥ 18.2 MΩ and total organic carbon ≤ 4 ppb).

### 5.2. Cultivation and Treatment

To start the experiment, refrigerated strawberry plants (*Fragaria* var. *Elsanta*) were obtained from Agra Cleassens (Agra, India). These one-year-old plants are harvested at the end of the harvesting year, preserving the roots in a fridge during winter. At the start of spring, the plants are put back into soil culture for strawberry production. Overall, these plants are known to produce more strawberries compared to year-zero plants. For our application, the presence of the roots on these plants was ideal to initiate the hydroculture and skip the sowing stage immediately.

Strawberry plants were grown in a Mammoth Lite + 45 type (450 × 450 × 1200 mm) culture chamber where the light (around 166 μmol m^−2^ s^−1^ for 12 h/day), temperature (24.6 ± 1.6 °C), and relative humidity were kept constant. The plants were grown in pure Hoagland solution using amber glass bottles until fruit development. Hoagland solution was made based on the original protocol [68]. Due to the novelty of this approach, the level of Hoagland solution in the bottles had to be optimized for successful cultivation. Therefore, the roots of 74 strawberry plants were only submerged up to half of the root length, while the complete root was submerged for another 74 plants. The experiment was performed twice over two years. In the first year, 24 plants were submitted to each condition. In the second year, the number of plants was expanded to 50 plants per condition.

Only the plants with the half-submerged roots were able to survive the initial stage of the experiment. Specifically, 54 plants were used in the next stage of the experiment. The plants were separated into three groups, the blank control, the natural MC mix (NM), and MC-LR. The blank control plants were grown in pure Hoagland solution. The NM plants were grown in Hoagland solution fortified with 100 µg L^−1^ total analyzed MCs by adding freeze-dried cyanobacterial mass. This biomass was collected from a Belgian pond during bloom. The freeze-dried by mass contained 207.41 µg g^−1^ total analyzed MCs, including MC-LR (38.1%), MC-RR (35.8%), MC-LA, MC-LF, MC-LY, MC-LW, MC-YR (20.3%), and MC-WR (4.3%). For the MC-LR condition, plants were grown in Hoagland solution fortified at 100 µg L^−1^ using pure MC-LR standard at 1 mg mL^−1^ obtained from ENZO Life Sciences. The Hoagland solution was made uniformly as earlier described [68]. Thereafter, MC-LR or the freeze-dried material was added to the solution. All solutions, the blank control, NM, and MC-LR, were horizontally shaken for 1 h to ensure a homogenous distribution of the toxins in the solution.

At the start of the experiment, 150 mL of Hoagland (blank control), NM Hoagland, or MC-LR Hoagland solution was added to the plants in the respective conditions. Thereafter, the solutions were added according to the needs of the plants, ensuring that the roots were not completely submerged. Every week, a new batch of Hoagland solution for each condition was prepared. A sample of 50 mL from each batch was stored to evaluate the MC(s) content at the end of the experiment. Samples were analyzed using a validated method [72].

All plants were grown until the fruits were ripe (after 5 weeks). The growth of one plant in the blank and one plant in the MC-LR condition stalled, resulting in a lack of fruit development. Both plants were removed from the experiment. The final numbers of sampled plants were consecutive, 17, 18, and 17, for the blank, NM, and MC-LR conditions.

Strawberries, roots, and greens (e.g., stems and leaves) were separated and ground. The samples were analyzed using a validated UHPLC-MS/MS method for MC quantification in fruits and vegetables [47].

### 5.3. Analytical Analysis

Overall, eight MCs (MC-RR, MC-LR, MC-YR, MC-LW, MC-WR, MC-LY, MC-LA, and MC-LF) in the Hoagland solution and the different strawberry plant parts were quantified using validated methods [47,71,72]. The Limits of Detection (LODs), Limits of Quantification (LOQs), and other validation parameters for the MCs were reported in these publications. In short, water samples were extracted using solid-phase extraction (SPE) (Agilent, Santa Clara, CA, USA, Bond Elute C18 cartridge, 6 mL, 500 mg, 40 µm) and eluted with 80% methanol. A Phenomenex RC syringe filter was consecutively used to filter impurities from the samples. The samples were stored at −20 °C until analysis [53]. On the other hand, the plant material samples were extracted with 80% methanol (4.5 mL) and sonicated for 15 min. After a 30 min incubation in an overhead shaker and subsequent centrifugation, the supernatants were evaporated to near dryness under a nitrogen stream. The remaining liquid (around 1.5 mL) was extracted using SPE (C18 cartridge) and filtered through a Phenomenex RC syringe filter. Samples could again be stored at −20 °C [47,71].

MC quantification in both the water and plant material was accomplished using UHPLC-MS/MS. Specifically, a Waters Acquity UPLC H-class was used, connected to a Xevo TQ-S (Waters, Milford, MA, USA). A Waters Acquity BEH C18 column (1.7 µm, 2.1 mm × 100 mm) was used to separate the toxins, as published earlier [18,47,54,55,71,72]. Fortified samples for every matrix were used as quality control during each analytical run. Moreover, matrix-matched calibration curves were used to quantify toxin concentrations.

### 5.4. Statistical Analysis

Average concentrations for MC-LR in the different sample types were calculated based on the combined data of the initial and validation experiments. Results defined as below LOQ (plant materials: 1 ng g^−1^ and water: 0.5 µg L^−1^) were excluded from these calculations. Boxplots were calculated and plotted using the R-studio Tidy verse package. Axis breaks were made with the ggbreak package [73]. The statistical analysis was also performed in R-studio 4.4.3. The normality and equal variance of the data were tested using the Shapiro–Wilk and F-test statistics, respectively. A combination of non-normally distributed data and unequal variance between treatment groups resulted in the use of the Wilcoxon test to calculate *p*-values (<0.05) for assessing the significant difference in toxin concentrations or weight for the different plant parts and/or growth conditions.

## Figures and Tables

**Figure 1 toxins-17-00285-f001:**
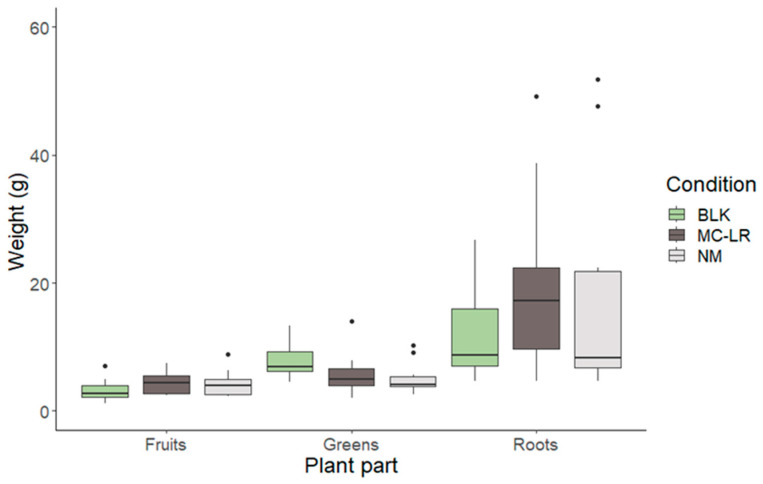
Distribution of the weights for all plant parts per irrigation condition, blank (BLK), fortified with MC-LR (MC-LR), or fortified with natural microcystin congener mix (NM). One outlier value (103.3 g) for the roots under the BLK condition was removed to increase the comprehension of the graph. However, the Wilcoxon test (*p* < 0.05) showed no significant difference between growth conditions for the same plant part, whether the outlier was excluded or included.

**Figure 2 toxins-17-00285-f002:**
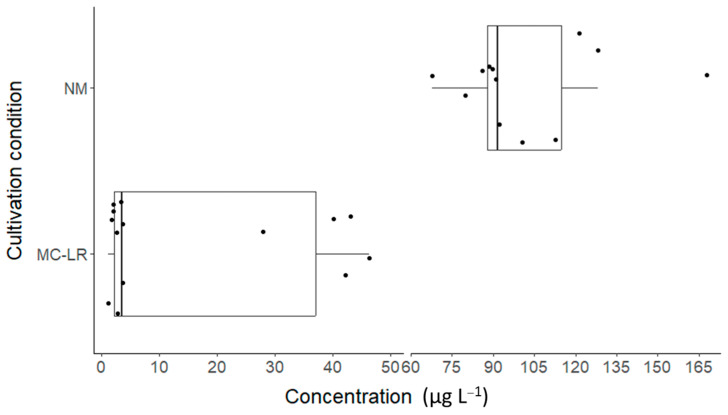
Boxplots showing the total measured microcystins of the natural microcystin congener mix in Hoagland solution (NM) and the MC-LR concentration in the MC-LR Hoagland solution. The dots indicate the individual data points.

**Figure 3 toxins-17-00285-f003:**
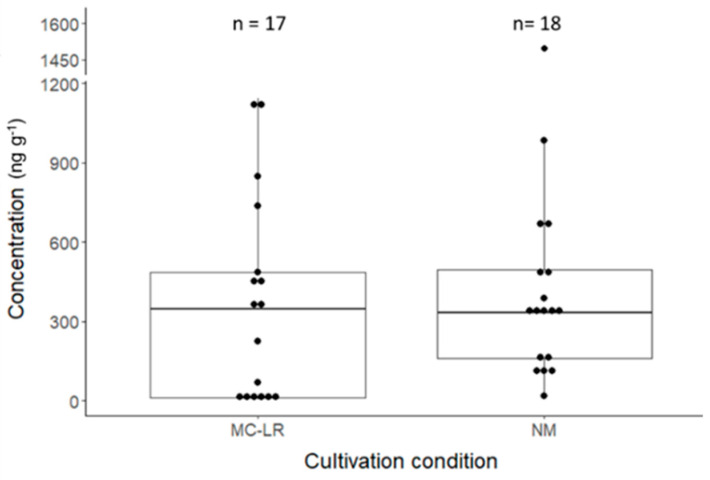
Boxplots showing the total measured microcystins or MC-LR concentration for roots treated with the natural microcystin congener mix in Hoagland solution (NM) or the MC-LR Hoagland solution, respectively. There was no significant difference between the concentration levels based on a Wilxocon test (*p* < 0.05). ‘n’ is the total number of samples in which MCs were detected.

**Figure 4 toxins-17-00285-f004:**
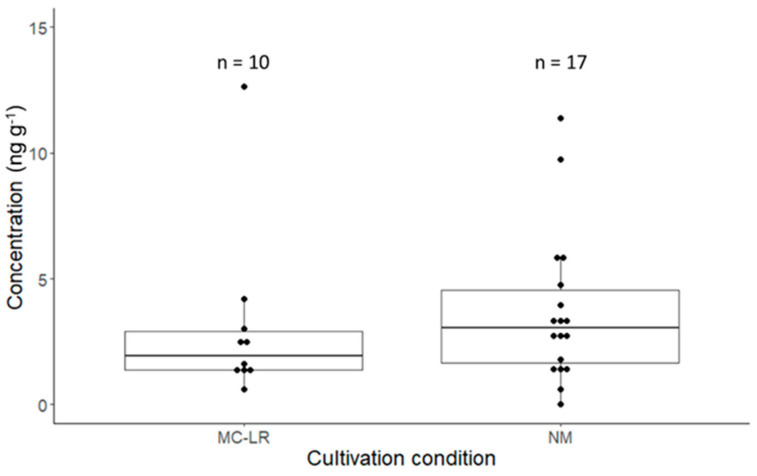
Boxplots showing the total measured microcystins or MC-LR concentration for greens treated with the natural microcystin congener mix in Hoagland solution (NM) or the MC-LR Hoagland solution, respectively. There was no significant difference between the concentration levels based on a Wilcoxon test (*p* < 0.05). ‘n’ is the total number of samples in which MCs were detected.

**Table 1 toxins-17-00285-t001:** Review of existing MC or MC-LR accumulation data in strawberry leaves and stems.

Cultivation Method	Toxin Concentration in Solution	Concentration (ng g^−1^)	Reference
Liquid hydroculture	100 µg L^−1^ MC-LR	2.8	current study
Liquid hydroculture	100 µg L^−1^ MCs	3.4	current study
Substrate hydroponic	20 µg L^−1^ MCs	6.59	[61]
Soil-based drip irrigation	100 µg L^−1^ MC-LR	3.4	[34]
Soil-based spray irrigation	100 µg L^−1^ MC-LR	5.6	[34]

## Data Availability

The original contributions presented in this study are included in the article/Appendix A. Further inquiries can be directed to the corresponding author.

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
