# Peer review of "Hydroculture Cultivation of Strawberries as Potential Reference Material for Microcystin Analysis: Approaches and Pitfalls"

_toxins, 2025, doi:10.3390/toxins17060285_

Round 1
Reviewer 1 Report
Comments and Suggestions for Authors
Comments to the author
- What specific physiological or biochemical barriers do you suspect are preventing the translocation of microcystins into the strawberry fruits, despite their accumulation in roots and greens?
- Could the observed absence of microcystins in the fruits be influenced by the timing of exposure during fruit ripening, and have you considered earlier or prolonged exposure periods?
- Given the hydrophilic nature of microcystins, did you evaluate the role of transport proteins or root membrane permeability in their uptake and distribution within the plant system?
- Were there any signs of stress or altered metabolic activity in the strawberry plants due to microcystin exposure, and how might this affect the generalizability of your model to other plant species?
- How reproducible and scalable is the hydroculture setup you developed, particularly for labs aiming to generate reference materials for regulatory or inter-laboratory validation studies?
- Considering the risk of root rot with full submersion, how do you foresee maintaining plant health over longer experimental periods while still ensuring effective toxin exposure?
- Did the use of cyanobacterial biomass (as opposed to pure microcystin-LR) introduce any complications in terms of consistency, additional metabolites, or microbial interactions that could affect uptake or detection?
- Have you identified or considered any other fruiting plant species that might be more suitable for use as reference material matrices for microcystin analysis, and what selection criteria would guide such a choice?

Author Response
Reviewer 1 :
Dear Reviewer, Thank you for your insightful comments. We hope we were able to provide clear answers where possible and justifications in the other cases. Textual changes are annotated in red and referred to in the responses by the lines in which they occur.
Comment 1: What specific physiological or biochemical barriers do you suspect are preventing the translocation of microcystins into the strawberry fruits, despite their accumulation in roots and greens?
Response 1: Thank you for this relevant question. This is still an unresolved question that can hopefully be answered using our setup in the future. As an analytical chemist, focused on toxin analysis and food safety, I’m not an expert on plant physiology and biochemistry. However, there are multiple possibilities. Micorcystin congeres might have difficulties crossing from the phloem to the fruits through the plasmodesmata. At this stage, there could be molecular regulation of the sieve elements, and the amphipathic nature of most microcystin congeners or local pH effects. On the other hand, higher glutathion levels in the fruits, compared to stems and roots, could interact and subsequently mask the microcystin congeners from our current analytical approach. Developing and utilizing HRMS or BMMP analysis could help elucidate this possibility. However, this was not possible in the framework of this project.
Comment 2: Could the observed absence of microcystins in the fruits be influenced by the timing of exposure during fruit ripening, and have you considered earlier or prolonged exposure periods?
Response 2: During our study, strawberry plants were exposed to microcystin congeners from the fruiting/ripening to the harvest of the fruits. Prolonging the exposure would require delaying the harvest for the fruits, reducing their quality and thus relevance as a reference material in a food context. However, starting the microsystin exposure before the fruiting stage (e.g., at the flowering stage) is possible. Yet, due to the cost of the pure microcystin-LR standard and limited available harvested cyanobacteria bloom biomass, this was not possible during this project. Longer exposure in literature was shown by Haida et al; 2023. As discussed in our manuscript, different methodological approaches prevent a reliable answer to your question.
Comment 3: Given the hydrophilic nature of microcystins, did you evaluate the role of transport proteins or root membrane permeability in their uptake and distribution within the plant system?
Response 3: We did not evaluate any molecular transport systems in the plant as this was out of scope for the project. Elucidating these systems would be very valuable but would require multiple years of research. We would be very happy to further collaborate with other research groups to investigate these mechanisms in the future.
Comment 4: Were there any signs of stress or altered metabolic activity in the strawberry plants due to microcystin exposure, and how might this affect the generalizability of your model to other plant species?
Response 4: We did not observe any signs of stress in the strawberry plants due to the microcystin exposure compared to the control. Analyzing metabolic activity was out of scope for this project.
Comment 5: How reproducible and scalable is the hydroculture setup you developed, particularly for labs aiming to generate reference materials for regulatory or inter-laboratory validation studies?
Response 5: As mentioned in lines 351 to 353, the shown results were obtained over 2 years, thus already replicating the experiment. Therefore, it should be feasible to replicate it multiple times. The setup should be able to scale up. If using our exact method, this is a question of space, in which we were very limited, and good control of the growth medium level. At a larger scale, this could be potentially also be accomplished with ebb and flow irrigation setups. We added a discussion on this topic in lines 284 to 293.
Comment 6: Considering the risk of root rot with full submersion, how do you foresee maintaining plant health over longer experimental periods while still ensuring effective toxin exposure?
Response 6: During the experiments, the roots were never completely dry. The root tip of the primary root and tertiary roots were always in contact with the solution, ensuring toxin exposure.
Comment 7: Did the use of cyanobacterial biomass (as opposed to pure microcystin-LR) introduce any complications in terms of consistency, additional metabolites, or microbial interactions that could affect uptake or detection?
Response 7: During the preparation of the biomass, it was freeze-dried. This should prevent any microbiological interaction with the microcystin congeners in the biomass mixture. All the biomass was homogenized and tested for homogeneity of microcystin congeners concentration within the different aliquots. With the exception of one outlier, all observed microcystin congeners concentrations within the Hoagland solution were close to the targeted 100 µg/L. We added some additional some additional information in the result section lines 123 to 127.
Furthermore, microbial interactions were not evaluated during the irrigation experiment. However, the stability of the microcystin congeners from this biomass was already shown in Van Hassel et al. 2024. Therefore, we do not suspect microbial degradation of the microcystin congeners. Due to the current liquid hydroculture setup, we do not expect rhizosome-bacteria-like interactions. On the other hand, we do not expect that additional metabolites will interfere with uptake of the microcystin congeners. The accumulated concentrations of pure MC-LR and the total MCs from the mix are very similar (e.g., not significantly different) for the roots and the greens, as shown in Figures 3 and 4. Our targeted UPLC-MS/MS method is validated and is not influenced by the presence of additional metabolites.
Comment 8: Have you identified or considered any other fruiting plant species that might be more suitable for use as reference material matrices for microcystin analysis, and what selection criteria would guide such a choice?
Response 8: The main choice will be driven by the ability to adapt our hydroculture setup for the selected fruiting plant. Fruit-bearing stalk plants, like tomato or cumcumber, would be an interesting choice as they consume a lot of water and have a great capability to produce roots and can be grown in hydroculture. Moreover, they are a relevant food source all over the world. However, berry plants, like the European blueberry would also be a good fit as they can also be grown in hydroculture and are commonly consumed. Of course, any crop plant that could be grown in hydroculture could be valuable to diversify the range of reference materials. Lines 299 to 311.

Reviewer 2 Report
Comments and Suggestions for Authors
Few concerns to be addressed.
The study attributes the low MC-LR recovery in Hoagland solution to degradation during storage at -20°C. However, freezing is generally considered a stable storage condition for many compounds. The claim lacks direct experimental evidence (stability tests at -20°C) or references to prior studies demonstrating MC-LR instability at this temperature.
In Section 2.2, the text states that minor MC congeners in roots had "concentrations on average lower than 1.5 µg g⁻¹," yet the total MC concentration in roots is reported as 417.1 ng/g (0.417 µg/g).
The absence of MCs in fruits is attributed to methodological differences (root vs. foliar exposure), but the paper does not explore biological barriers or translocation dynamics (phloem/xylem transport limitations). Comparisons to studies with longer exposure durations are mentioned but not critically analyzed, leaving a gap in explaining why fruits remained toxin-free.
The use of refrigerated, one-year-old strawberry plants (Fragaria var. Elsanta) may introduce physiological variability compared to fresh plants. The study does not address whether refrigeration affected toxin uptake or metabolism.
While the Hoagland solution’s MC-LR degradation is blamed on -20°C storage, plant extracts were similarly stored at -20°C. The stability of MCs in plant matrices under frozen conditions is not discussed.
The conclusion dismisses strawberries entirely as reference material (RM) due to uncontaminated fruits. However, roots and greens accumulated MCs and could still serve as RM for specific matrices.
Author Response
Reviewer 2:
Dear Reviewer, Thank you for your insightful comments. We hope we were able to provide clear answers where possible and justifications in the other cases. Textual changes are annotated in red and referred to in the responses by the lines in which they occur.
Comment 1: The study attributes the low MC-LR recovery in Hoagland solution to degradation during storage at -20°C. However, freezing is generally considered a stable storage condition for many compounds. The claim lacks direct experimental evidence (stability tests at -20°C) or references to prior studies demonstrating MC-LR instability at this temperature.
Response 1: Thank you for this comment. While we do think the low recovery are attributed to the longer storage in Hoagland solution due to it’s lower pH compared to water (5.4-6.2), we understand that this should be further evaluated before making such steadfast claims. Therefore, we will adjust this part of the discussion, broadening it towards the potential reasons for instability, including adsorption to the recipient. Lines 214 to 226.
Comment 2: In Section 2.2, the text states that minor MC congeners in roots had "concentrations on average lower than 1.5 µg g⁻¹," yet the total MC concentration in roots is reported as 417.1 ng/g (0.417 µg/g).
Response 2: Thank you for the comment. This was a typo. It should be 1.5 ng g⁻¹ The change was made in Line 165.
Comment 3: The absence of MCs in fruits is attributed to methodological differences (root vs. foliar exposure), but the paper does not explore biological barriers or translocation dynamics (phloem/xylem transport limitations). Comparisons to studies with longer exposure durations are mentioned but not critically analyzed, leaving a gap in explaining why fruits remained toxin-free.
Response 3: Thank you for this comment. We agree this is a major knowledge gap that should be further investigated, as it is not described in the limited available literature on the subject of cyanotoxin contamination of plants. However, this was not the scope of our project and would require multiple years of study to elucidate these mechanisms. However, our study does provide an early step towards elucidating these questions by working towards reliable analytical methods and developing reference material that could be used to validate toxin analysis during research. Lines 263 to 271
Comment 4: The use of refrigerated, one-year-old strawberry plants (Fragaria var. Elsanta) may introduce physiological variability compared to fresh plants. The study does not address whether refrigeration affected toxin uptake or metabolism.
Response 4: Thank you for this comment. The use of fresh plants was not discussed in the manuscript. However, we initially tried in a pretrial phase to transfer 30 fresh strawberry plants from a soil to a hydroculture setup before the flowering stage. However, none of the plants survived the transfer. Therefore, they could not be exposed to the toxins and a comparison between uptake and metabolism between refrigerated and fresh plants was impossible. We choose not to include this pretrial in the manuscript to streamline our message and not overburden the readers. We do want to emphasize that one-year-old plants only consist of limited roots at the beginning of the experiment. They grew fresh roots, stems, and leaves during the experiments.
Comment 5: While the Hoagland solution’s MC-LR degradation is blamed on -20°C storage, plant extracts were similarly stored at -20°C. The stability of MCs in plant matrices under frozen conditions is not discussed.
Response 5: For the stability of MC-LR in plant matrix, we do not expect any problems. Plant tissues are generally analyzed within two weeks after harvest and grinding, as sufficient samples can be obtained to merit an analytical run. This was not the case for the Hoagland solution. Moreover, we have already multiple years of experience analyzing microcystins in plant matrices. For our paper, Van Hassel et al. 2023, we reanalyzed a selection of basil samples throughout a period of 2 years and shared the samples with collaborators in Canada and Norway for analysis. Results of these analyses all provided similar concentrations of MC-LR.
Comment 6: The conclusion dismisses strawberries entirely as reference material (RM) due to uncontaminated fruits. However, roots and greens accumulated MCs and could still serve as RM for specific matrices.
Response 6: We have rephrased this part of the discussion and conclusion. Lines 284 to 293, 322 to 323

Reviewer 3 Report
Comments and Suggestions for Authors
- The methods section states that a Wilcoxon test calculates p-values (<0.05) to assess significant differences between plant parts or growth conditions. Yet, the shown figures and tables don't display these results.
- The statement “A validated method quantified microcystin congeners in the Hoagland solution and different strawberry plant parts.” is the method from your laboratory’s previous research? What are the method’s accuracy and stability? The methods in the two cited papers are for water environments, not complex plant matrices. Matrix effects can greatly impact results, so I believe the toxin detection method for plant matrices needs more validation.
- In Figure 2, why do samples in the NM group show concentrations exceeding 100 µg/L? Toxin concentrations in the solution should decrease after absorption.
- Figures 3 and 4 have different sample sizes. According to the methods section, the final numbers of sampled plants were 17, 18, and 17 for the blank, NM, and MC-LR conditions. Some data are missing in Figure 4.
Author Response
Review 3
Dear Reviewer, Thank you for your insightful comments. We hope we were able to provide clear answers where possible and justifications in the other cases. Textual changes are annotated in red and referred to in the responses by the lines in which they occur.
Comment 1: The methods section states that a Wilcoxon test calculates p-values (<0.05) to assess significant differences between plant parts or growth conditions. Yet, the shown figures and tables don't display these results.
Response 1: We have clarified this more appropriately in the method section (lines 414). For all relevant Figures (1, 3,4), the following statement had already been included in their description: “There is no significant difference between the concentration levels based on a Wilxocon test (p< 0.05).” Showing this in the figure would confuse the readers, as usually only significant differences are annotated in figures.
Comment 2: The statement “A validated method quantified microcystin congeners in the Hoagland solution and different strawberry plant parts.” is the method from your laboratory’s previous research? What are the method’s accuracy and stability? The methods in the two cited papers are for water environments, not complex plant matrices. Matrix effects can greatly impact results, so I believe the toxin detection method for plant matrices needs more validation.
Response 2: Thank you for your comment, we are glad that there are researches as concerned about proper methodology and matrix effect as we are.
.”. Indeed, reference 72 (previously 65) reference to our analytical method for MCs in water matrix: ‘Van Hassel, W.H.R.; Huybrechts, B.; Masquelier, J.; Wilmotte, A.; Andjelkovic, M. Development, Validation and Application of a Targeted LC-MS Method for Quantification of Microcystins and Nodularin: Towards a Better Characterization of Drinking Water. Water 2022, 14, 1195, doi:10.3390/w14081195.”
However, the reference 47 refers to our paper “Towards a Better Quantification of Cyanotoxins in Fruits and Vegetables: Validation and Application of an UHPLC-MS/MS-Based Method on Belgian Products. Separations 2022, 9, 319, doi:10.3390/separations9100319
In this publication, we fully describe our validated method for strawberries, carrot and lettuce in detail, following the European Commission 2002/657/EC decision to implement Council Directive 96/23/EC concerning the performance of analytical methods and the interpretation of results. As an ISO17025-accredited lab. We also perform third-line tests (due to a lack of PT programs) at the appropriate times to ensure the stability of the method. We stopped mentioning this in our papers, as reviewers usually find this information unnecessary.
As an amendment to the paper, we have added a reference to our equally validated method in Basil. (Van Hassel, W.H.R.; Abdallah, M.F.; Gracia Guzman Velasquez, M.; Miles, C.O.; Samdal, I.A.; Masquelier, J.; Rajkovic, A. Experimental Accumulation and Depuration Kinetics and Natural Occurrence of Microcystin-LR in Basil (Ocimum Basilicum L.). Environ. Pollut. 2024, 347, 123715, doi:10.1016/j.envpol.2024.123715)
Furthermore, there was a faulty reference in line 319.
Moreover, the measurement uncertainty of our method was also mentioned in lines 123,141, 145 and 146)
As you can also read in these publications, matrix effects were assessed during validation for the different matrices and we use a matrix-matched calibration curve during analysis to mediate matrix effects, also mentioned in line 402 and 403.
Comment 3: In Figure 2, why do samples in the NM group show concentrations exceeding 100 µg/L? Toxin concentrations in the solution should decrease after absorption.
Response 3: These are results measured in the Hoagland solution before addition to the plants. Therefore, adsorption of the toxins is not a factor. Moreover, our previous research in basil shows that the concentration of the MCs actually remains stable for one week in the solution. Therefore, we hypothesized that the available toxins are absorbed at the same rate as the water/growth solution as the whole is consumed by the plant.
Concerning the higher concentrations, we mentioned in line 122. Part of this variability can be explained by the measurement uncertainty of our method. Moreover, the freeze-dried cyanobacteria material is most likely not completely homogeneous. The average concentration of the toxins in the biomass was used to calculate the weight of the biomass added to the Hoagland solution. could result in some samples having higher concentrations. Yet, overall, the average exposed concentration was 100.41 µg L-1, nicely approaching the targeted value. We added this information to the result section from lines 123 to 127.
Comment 4: Figures 3 and 4 have different sample sizes. According to the methods section, the final numbers of sampled plants were 17, 18, and 17 for the blank, NM, and MC-LR conditions. Some data are missing in Figure 4.
Response 4: ‘n’ shows the number of samples in which toxins were detected. We agree that this should be addressed in the figure description. This was amended in lines 158 and 180.

Reviewer 4 Report
Comments and Suggestions for Authors
The manuscript presents a relevant methodological approach. However, the expression and interpretation of the results require significant improvement to meet publication standards.
A major concern is the limited clarity and specificity regarding the microcystin (MC) congeners analyzed. Although the manuscript refers broadly to "microcystins" or "MC congeners," it fails to clearly specify which congeners were detected or quantified in the results section (figure 1, 2 , 3). This lack of detail undermines the interpretability and scientific value of the findings, as different MC congeners possess distinct physicochemical properties and toxicokinetic behaviors that can influence their bioavailability, persistence, and toxic effects.
The authors should identify and quantify the individual MC congeners observed in their study. Furthermore, the discussion should be expanded to critically evaluate these results in the context of the known differences among congeners, particularly with respect to parameters such as hydrophobicity, cellular uptake, and metabolic stability.
Additionally, while the methodology is described as promising in the conclusions, the manuscript would benefit from a dedicated discussion on future perspectives. This should include the potential for broader application, current limitations of the method, and recommendations for further validation or refinement.
Author Response
Review 4
Dear Reviewer, Thank you for your insightful comments. We hope we were able to provide clear answers where possible and justifications in the other cases. Textual changes are annotated in red and referred to in the responses by the lines in which they occur.
Comment 1: A major concern is the limited clarity and specificity regarding the microcystin (MC) congeners analyzed. Although the manuscript refers broadly to "microcystins" or "MC congeners," it fails to clearly specify which congeners were detected or quantified in the results section (figure 1, 2 , 3). This lack of detail undermines the interpretability and scientific value of the findings, as different MC congeners possess distinct physicochemical properties and toxicokinetic behaviors that can influence their bioavailability, persistence, and toxic effects.
Resonse 1: We apologize if we misunderstood this comment but we do not agree with it. Figure 1 shows the weights of the plants. Not the accumulated MCs. For Figures 2,3 and 4 we agree that we only show either MCs or MC-LR, depending on the growth condition. As we recognize the value of identifying the different congeners separately, we already added accompanying figures in the supplementary files ( as mentioned in lines 130-131, 160-161 and 172-173). In these figures, we compare the spread of each detected microcystin congener in the samples compared to the sum of all analyzed MCs. We chose not to add these figures directly to the manuscript to prevent overloading it with figures. For further clarification, we will add the result tables to the supplementary files and mention them in the manuscript in lines 131 to 132, 161 to 162 and 173 to 174. Moreover, we have listed the congeners, included in our analytical method, in Lines 385 and 386.
Comment 2: The authors should identify and quantify the individual MC congeners observed in their study. Furthermore, the discussion should be expanded to critically evaluate these results in the context of the known differences among congeners, particularly with respect to parameters such as hydrophobicity, cellular uptake, and metabolic stability.
Response 2: As mentioned in our answer to the first comment, we did detect and quantify the individual congeners present in the plant tissues.
We recognize the importance of the structural diversity of the different microcystins and that the influence of their specific characteristics might influence their cellular uptake and metabolic activity. However, on this subject, there is a significant knowledge gap with no research combining a proper plant model and reliable quantification of the different MCs elucidating the effect of independent microcystin congeners on accumulation in the plant and their metabolic stability. We added this sentiment to the discussion in lines 263 to 271
Comment 3: Additionally, while the methodology is described as promising in the conclusions, the manuscript would benefit from a dedicated discussion on future perspectives. This should include the potential for broader application, current limitations of the method, and recommendations for further validation or refinement.
Response 3: The requested discussion was added in Lines 284to 311

Round 2
Reviewer 2 Report
Comments and Suggestions for Authors
The manuscript is improved and can be accepted for publication
Reviewer 3 Report
Comments and Suggestions for Authors
No more comments
Reviewer 4 Report
Comments and Suggestions for Authors
All my concerns have been answered correctly